# Similarity and dissimilarity in alterations of the gene expression profile associated with inhalational anesthesia between sevoflurane and desflurane

Takehiro Nogi[1], Kousuke Uranishi[2], Ayumu Suzuki[2], Masataka Hirasaki[3], Tina Nakamura[1], Tomiei Kazama[1], Hiroshi Nagasaka[1], Akihiko Okuda[2]*, Tsutomu Mieda[1]*

1 Department of Anesthesiology, Saitama Medical University Hospital, Morohongo, Moroyama, Iruma-gun, Saitama, Japan, 2 Division of Biomedical Sciences, Research Center for Genomic Medicine, Saitama Medical University, Yamane, Hidaka Saitama, Japan, 3 Department of Clinical Cancer Genomics, Saitama Medical University International Medical Center, Yamane, Hidaka, Saitama, Japan

* akiokuda@saitama-med.ac.jp (AO); tmieda@saitama-med.ac.jp (TM)

**Data Availability Statement:** RNA sequence data used for this study were deposited in Gene

## Abstract

Although sevoflurane is one of the most commonly used inhalational anesthetic agents, the popularity of desflurane is increasing to a level similar to that of sevoflurane. Inhalational anesthesia generally activates and represses the expression of genes related to xenobiotic metabolism and immune response, respectively. However, there has been no comprehensive comparison of the effects of sevoflurane and desflurane on the expression of these genes. Thus, we used a next-generation sequencing method to compare alterations in the global gene expression profiles in the livers of rats subjected to inhalational anesthesia by sevoflurane or desflurane. Our bioinformatics analyses revealed that sevoflurane and, to a greater extent, desflurane significantly activated genes related to xenobiotic metabolism. Our analyses also revealed that both anesthetic agents, especially sevoflurane, downregulated many genes related to immune response.

## Introduction

Inhalational anesthesia using halogenated anesthetic agents is a common method of inducing general anesthesia [1,2]. Sevoflurane is currently one of the most commonly used inhalational anesthetic agents because of its numerous beneficial characteristics in the clinic [3,4]. For example, sevoflurane is extremely refractory to being dissolved in the blood, leading to its elimination from the lung as vapor [5,6]. Furthermore, sevoflurane, unlike other halogenated anesthetic agents, does not produce trifluoroacetic acid, which induces severe hepatic injury [7,8]. Despite this, desflurane, another halogenated anesthetic agent, is rapidly gaining popularity, and the prevalence of inhalational anesthesia using desflurane is approaching that of sevoflurane. Although desflurane has some disadvantages, including causing respiratory tract irritation [9–13], an obvious advantage of its clinical use is the relatively rapid induction of an anesthetic state upon its administration and the rapid recovery of patients from that state after the cessation of its supply compared with other anesthetic agents [14–19]. These effects of desflurane are

Expression Omnibus-NCBI under accession number GSE244436.

**Funding:** Takehiro Nogi is the recipient of in-house grants from Saitama Medical University (Internal Grants, 21-B-1-07 and 23-B-1-15) and Saitama Medical University Hospital (02-E-1-08). Akihiko Okuda is the recipient of a grant from the Japan Society for the Promotion of Science (KAKENHI: grant number 23H02678). However, these funders had no role in study design, data collection and analysis, decision to publish, or preparation of the manuscript.

**Competing interests:** The authors have declared that no competing interests exist.

related to its marked low solubility in the blood, which is lower than that of sevoflurane [6,20,21]. An additional notable characteristic of desflurane is its extreme resistance to degradation and biotransformation [6,22–24], further elevating its safety level for clinical use. Thus, sevoflurane and desflurane have beneficial features as inhalational anesthetic agents.

To date, several studies have demonstrated that inhalation anesthesia is associated with the transcriptional activation of genes related to xenobiotic metabolism [25,26]. In addition, inhalation anesthesia reduces the expression of immune response-related genes [27–29]. However, whether these anesthetic agents have different influences on these genes is unknown. In this study, we addressed this question by comparing alterations in the global gene expression profiles of livers from rats subjected to inhalational anesthesia with desflurane or sevoflurane. Consistent with a previous report, we found that inducing general anesthesia with sevoflurane led to significant activation of genes related to xenobiotic metabolizing enzymes [25,26]. We also found that desflurane activated these genes more prominently than sevoflurane. Furthermore, our data revealed that sevoflurane and desflurane downregulated many genes related to immune response, and that the repressive effect of sevoflurane on these genes was more profound than that of desflurane.

## Materials and methods

### Animal experiments

Male Wistar rats (6 weeks old, 140–160 g body weight) purchased from Japan SLC Inc. (Hamamatsu, Japan) were housed in plastic cages with free access to food and water at 23˚C under controlled lighting (12:12-hour light/dark cycle: lights on at 7:00 AM) for at least 1 week to acclimatize. Rats were deprived of food and water for 2 hours prior to experimentation and then subjected to inhalational anesthesia via nose cones using sevoflurane (4.5% gas-air mixture) [25] or desflurane (6.0% gas-air mixture) [23] with a 3 L/min flow of 50% oxygen. During anesthesia, rats were allowed to breathe spontaneously, and the value of saturation of percutaneous oxgen measured using a pulse oximeter through the lower extremities of rats was strictly controlled so as not to drop below 98%. In addition, the body temperatures of anesthetized rats were maintained at 36.5–37.5˚C. After 6 hours, the rats subjected to inhalational anesthesia with sevoflurane or desflurane were sacrificed by decapitation, and the left lateral lobe of the liver was quickly isolated from each rat. Then, a portion of each liver was immersed in RNA later after rinsing with phosphate-buffered saline, and stored at 4˚C. For control rat livers, rats subjected to inhalational anesthesia with sevoflurane or desflurane were immediately sacrificed after the loss of consciousness, and liver specimens were prepared and stored as described above. All rats, including those used as controls, were sacrificed at approximately 3:00 PM to avoid the effect of differences in the circadian rhythm phase on gene expression. The protocol for these experiments was approved by the Institutional Review Board on the Ethics of Saitama Medical University (permission numbers 3301, 3547, and 3796).

### RNA preparation and reverse transcription

Total RNAs were prepared from the livers of rats subjected to inhalational anesthesia with sevoflurane or desflurane for 6 hours or less than 1 minute using an RNeasy Midi Kit (Qiagen, Venlo, Netherlands) according to the manufacturer's instructions. RNAs were then used to obtain cDNAs by reverse transcription as described previously [30].

### Quantitative PCR

cDNAs obtained by reverse transcription were used for quantitative PCR (qPCR) using the following TaqMan probes. *Cyp2b1*: Rn01457880_m1; *Por*: Rn00580820_m1; *Alas1*:

Rn00577936_m1; *Irf1*: Rn01456791_m1; *Mx2*: Rn01444341_m1; *Ccl6*: Rn01456400_m1; *Il33*: Rn01759835_m1; and *Gapdh*: Rn01775763_g1. qPCR was performed in triplicate using livers from 12 rats (3 rat livers for each condition), and the results were normalized to *Gapdh* expression levels.

## RNA sequencing

The integrity of total RNA was checked using 4200 TapeStation (Agilent Technologies) before generating the library. Libraries were prepared with total RNAs from four samples (N = 1 for each condition) using a Stranded Total RNA Prep, Ligation with Ribo-Zero Plus Kit (Illumina, San Diego, CA) according to the manufacturer's instructions. RNA sequencing was performed on a NovoSeq 6000 system (Illumina, Albany, NY), by paired-end 101 bp reads, with 40–60 M reads for each sample. Sequence reads were trimmed to remove low-quality sequences and adapter sequences using sickle 1.33 (parameter -q 30 -l 20). Trimmed reads were then mapped to the rn6 reference genome using HISAT2 (version 2.1.0) with default parameters. The mapped reads were sorted using SAMtools (version 1.10). After removing small RNA genes whose lengths were equal to or shorter than 200 base pairs from the gene list of the Genomic Annotation Resource (https://hgdownload.soe.ucsc.edu/goldenPath/rn6/bigZips/genes/rn6.refGene.gtf.gz), the gene transfer format was used for read count extraction and normalization by StringTie (version 2.1.2). To identify genes activated by sevoflurane or desflurane, genes whose values of transcripts per kilobase million (TPM) were higher than two in the anesthetized state were selected from the list. Then, genes whose TPM values in the anesthetized state were more than 2-fold higher than those obtained from respective control rats were selected. Likewise, genes whose TPM values were higher than two in the control state were used as the starting list to identify genes repressed by anesthesia.

## Gene ontology and gene set enrichment analyses

Gene Ontology (GO) analysis was performed using DAVID web tools (http://david.abcc.ncifcrf.gov). Gene Set Enrichment Analysis (GSEA) [31] was conducted according to the method described on the GSEA homepage (http://www.gsea-msigdb.org/gsea/index.jsp) using three different platforms of gene sets, "biological process of Gene Ontology", "Kyoto Encyclopedia of Genes and Genome", and "Reactome Pathway Database".

## Statistical analysis

All data from qPCR were subjected to the Student's *t*-test (two-tailed) to examine statistical significance. The following marks were used to indicate the extent of statistical significance: ***, $P<0.001$; **, $P<0.01$; *, $P<0.05$; NS (not significant), $P>0.05$.

## Results

### Genome-wide expression analyses of livers from rats subjected to inhalational anesthesia

We conducted comprehensive gene expression analyses using next-generation sequencing to compare alterations in the global expression profiles of rat livers caused by the inhalational anesthetic agents sevoflurane and desflurane (S1 Fig). Our data revealed that 201 and 282 genes were transcriptionally activated more than 2-fold by sevoflurane and desflurane, respectively, in which 59 genes were commonly activated by both anesthetic agents (Fig 1A, S1 Table). Next, these upregulated gene sets were assigned to GO classification to correlate gene expression changes with overall molecular functions. These analyses yielded 3 and 14 specific

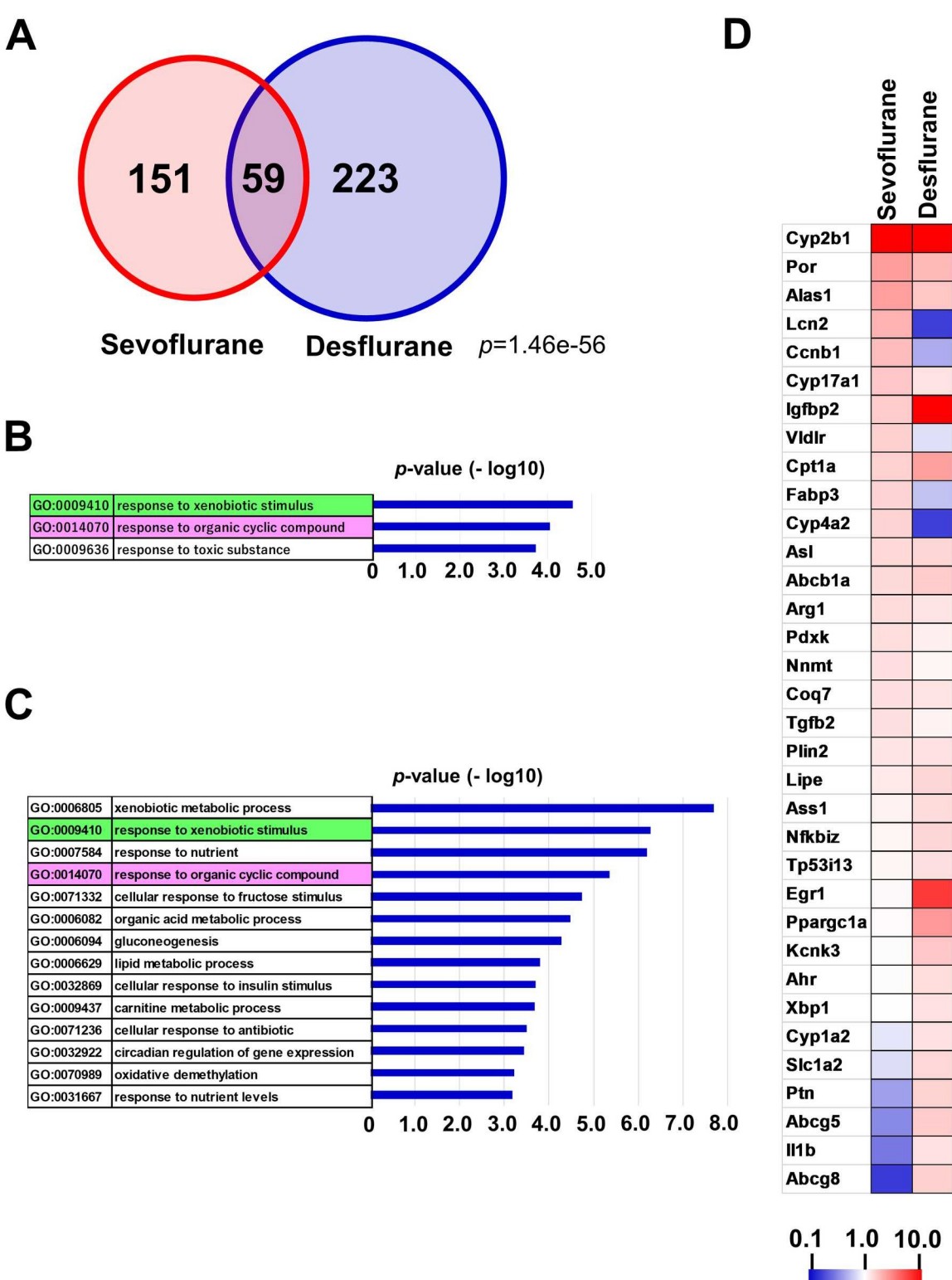

**Fig 1. GO analyses of genes upregulated by inhalational anesthesia.** (A) A Venn diagram showing a comparison of genes upregulated more than 2-fold by sevoflurane or desflurane. *P*-value for the significance of the overlap between two gene sets was calculated by a hypergeometric test. Lists of these genes are provided using their official gene symbols in S1 Table. (B, C) Genes whose expression was activated more than 2-fold by sevoflurane (B) or desflurane (C) were individually subjected to GO analyses using DAVID web tools (http://david.abcc.ncifcrf.gov). GO terms with a *p*-value less than $10^{-3}$ were selected and subjected to AmiGo2 analyses (http://amigo.

geneontology.org/amigo/landing) to eliminate synonymous terms. GO terms identical between sevoflurane and desflurane treatments were marked in distinct colors (green and pink). (D) Heatmap showing a comparison of altered expression levels of xenobiotic metabolism-related genes activated by sevoflurane and/or desflurane. Genes that contributed to the identification of the GO term "response to xenobiotic stimulus (0009410)" by sevoflurane treatment or desflurane treatment were combined to generate a gene list of the heatmap.

GO terms related to sevoflurane (Fig 1B) and desflurane (Fig 1C), respectively, whose *p*-values were less than $10^{-3}$. As expected, two GO terms related to the response to drug treatment (marked in green and pink) were obtained by sevoflurane and desflurane treatments, although desflurane had a higher statistical significance than sevoflurane. One-on-one comparisons with a heat map revealed that genes activated by inhalation anesthesia among the members constituting the GO term "response to xenobiotic stimulus" did not overlap much between the sevoflurane and desflurane groups (Fig 1D). However, unexpectedly, regression analysis suggested a high correlation (R2 = 0.9371) between these two groups (S2A Fig, upper panel), even though a substantial number of genes were activated specifically by sevoflurane or desflurane. As a possible explanation for this apparent discrepancy, we considered that the profound activation of the *Cyp2b1* gene by both anesthetic agents may skew the proper evaluation of the data. Consistent with this notion, the analysis demonstrated no apparent correlation between these two groups in cases where data related to *Cyp2b1* were removed as an outlier from the gene list (S2A Fig, lower panel).

We also conducted analyses for genes downregulated by anesthetic agent treatment (Figs 2A and S1, S1 Table). Compared with the upregulated genes, genes downregulated more than 2-fold than their respective controls showed less intensive overlap between the two anesthetic agents. GO analyses of these downregulated gene sets yielded 17 and 5 specific GO terms related to sevoflurane (Fig 2B) and desflurane (Fig 2C), respectively, whose *p*-values were less than $10^{-3}$. Consistent with the less intensive overlap between sevoflurane and desflurane with respect to downregulated genes, no common GO term was obtained. Notably, many of the GO terms associated with sevoflurane were related to immunological reaction (indicated by red letters). Because this finding most probably reflects the repression of immune response-related genes in blood cells, such as lymphocytes that were included in the liver samples, these results indicate that immunological response may be impaired by sevoflurane treatment. Unlike sevoflurane treatment, no apparent biological relatedness was evident among five GO terms associated with desflurane treatment, none of which were related to immune response, suggesting that desflurane may exert no, or at least less, significant repression of genes related to immune response compared with sevoflurane. Unexpectedly, the same GO term, "response to xenobiotic stimuli" (GO:0009410) obtained for genes activated by desflurane and sevoflurane, was also obtained in the analyses of genes downregulated by desflurane, suggesting that inhalation anesthesia with desflurane induces more complex responses in the liver than originally anticipated. Although GO analyses did not provide any indication of the alleviation of immunological responses by desflurane treatment, we manually inspected our RNA sequence data to determine whether expression levels of immunological response-related genes were not affected by desflurane. First, we used a gene set in which genes downregulated more than 2-fold by sevoflurane treatment were selected from among the members of the GO term "defense response to virus" (GO:0051607) for the analysis. A heatmap visualization of gene expression revealed that none of these genes were noticeably activated by desflurane, but many showed reduced expression levels, although the magnitude of downregulation was much less significant compared with that induced by sevoflurane (Fig 2D). Likewise, many genes that contributed to the GO terms "immune response" (GO:0006955) and/or "response to bacterium" (GO:0009617) as sevoflurane treatment-specific terms also had a tendency to be

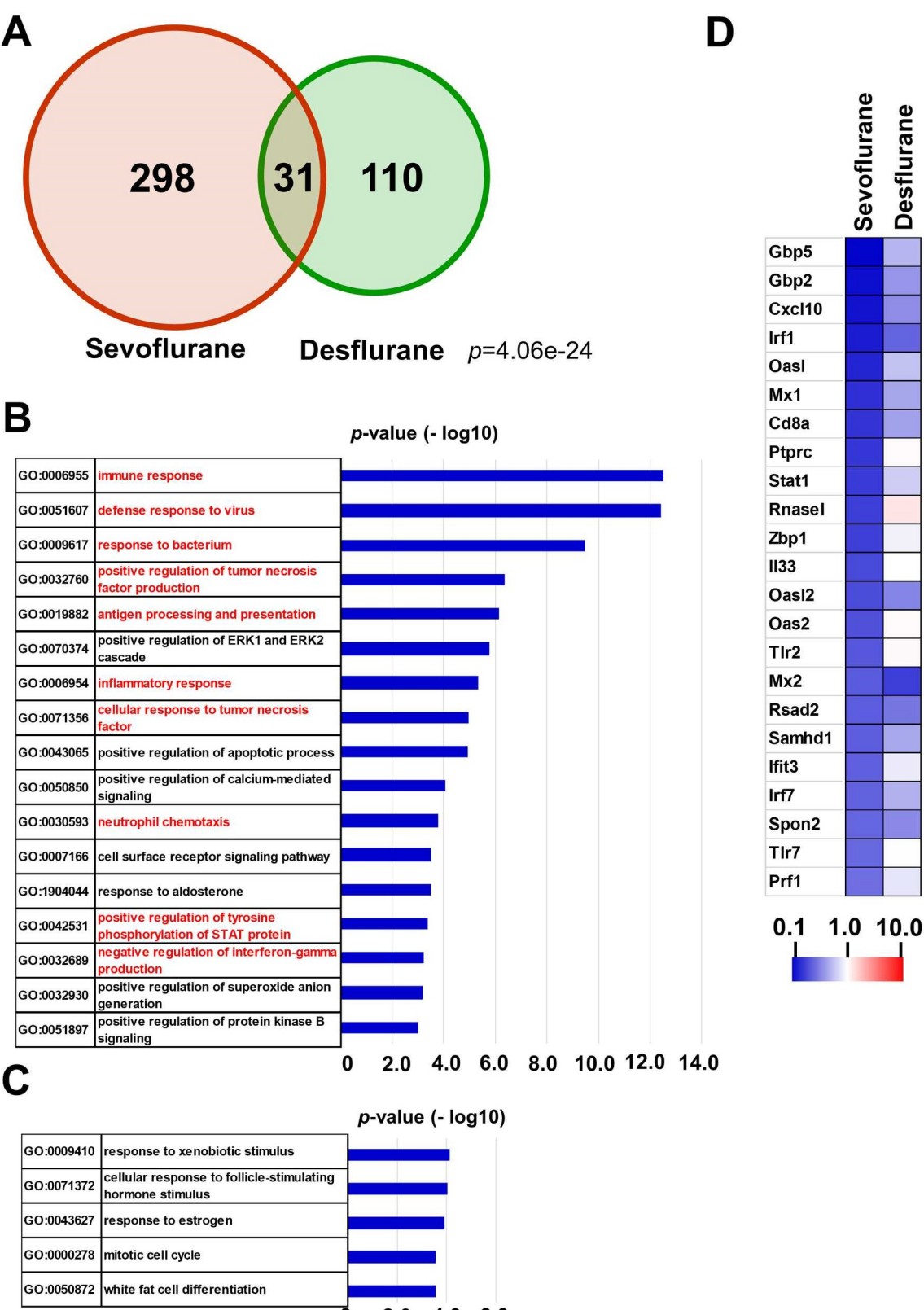

**Fig 2. GO analyses of genes downregulated by inhalational anesthesia.** (A) A Venn diagram showing a comparison of genes downregulated more than 2-fold by sevoflurane or desflurane. *P*-value for the significance of the overlap between two gene sets was

calculated by a hypergeometric test. Lists of these genes are provided using their official gene symbols in S1 Table. (B, C) Genes whose expression was downregulated more than 2-fold by sevoflurane (B) or desflurane (C) were individually subjected to GO analyses as in Fig 1B and 1C. GO terms related to immune response in (B) are indicated in red. (D) Heatmap showing a comparison of alterations in the expression levels of immune response-related genes by sevoflurane (left column) and desflurane (right column). Genes that contributed to the identification of the GO term "defense response to virus (0051607)" as a sevoflurane treatment-specific term were used for the analyses.

downregulated by desflurane (S3 Fig). These data indicate that the repressing effect of immune response-related genes by inhalation anesthesia was not specific to sevoflurane, but the magnitude of the repressive effect was greater for sevoflurane than desflurane. Although the manual inspection of our RNA-sequence data indicated that sevoflurane as well as desflurane led to the repression of immunological response-related genes, regression analyses revealed that there were no (GO:0051607 and GO:0006955) (S2B and S3A Figs) or weak (GO:0009617) (S3B Fig) correlations in expression changes between sevoflurane and desflurane treatments. These data indicated that, with a few exceptions, genes that were downregulated strongly or weakly by sevoflurane were not downregulated strongly and weakly by desflurane, suggesting that desflurane is not simply an anesthetic agent that downregulates immune-related genes more weakly than sevoflurane.

## Identification of gene sets coordinately regulated by desflurane and/or sevoflurane via gene set enrichment analysis

In addition to the above GO analyses, we also conducted GSEA to assess similarities and differences in phenotypic changes that occurred in rat livers, including blood cells subjected to inhalational anesthesia by sevoflurane or desflurane (S3 Fig). In the analyses, we used three publicly available databases, "biological process of Gene Ontology", "Kyoto Encyclopedia of Genes and Genome", and "Reactome Pathway Database". First, we found that three gene sets related to xenobiotic metabolism including "DRUG_METABOLISM_CYTOCHROME_P450" were identified as gene sets activated specifically by desflurane treatment (S3A Fig). Of note, none of these gene sets were identified by GSEA using RNAs from the livers of rats in the sevoflurane group, even though such terms were identified by GO analyses. These results indicate that xenobiotic metabolism-related genes may not be as coordinately regulated by sevoflurane compared with desflurane. This notion is consistent with the data shown in Fig 1A and 1B where GO terms related to xenobiotic metabolism were statistically more significant in the desflurane group compared with the sevoflurane group. We also found that numerous terms related to immunological reactions were identified in the gene sets downregulated by sevoflurane and desflurane. Of note, some terms, including "CYTOKINE_CYTOKINE_RECEPTOR_INTERACTION" were commonly identified in both groups. However, the effect of sevoflurane treatment appeared to be more profound than desflurane treatment on the basis of the total number of identified terms related to immunological reactions (43 and 15 terms for sevoflurane and desflurane treatments, respectively) and normalized enriched score (NES) (number of terms whose NES values were lower than −2.0 = 31 and 1 for sevoflurane and desflurane treatments, respectively). These findings were consistent with the GO analysis data shown in Fig 2B and 2C, where many and no terms related to immune response were obtained for the sevoflurane and desflurane-treated rat livers, respectively. Fig 3 shows representative snapshots of GSEA showing a tendency for the positive regulation of genes constituting the term "DRUG_METABOLISM_CYTOCHROME_P450" by desflurane treatment (Fig 3A) and the negative regulation of genes constituting the term "ADAPTIVE_IMMUNE_RESPONSE" by sevoflurane treatment (Fig 3B). In addition, Fig 3C shows snapshots of GSEA showing a tendency for the negative regulation of genes constituting the term

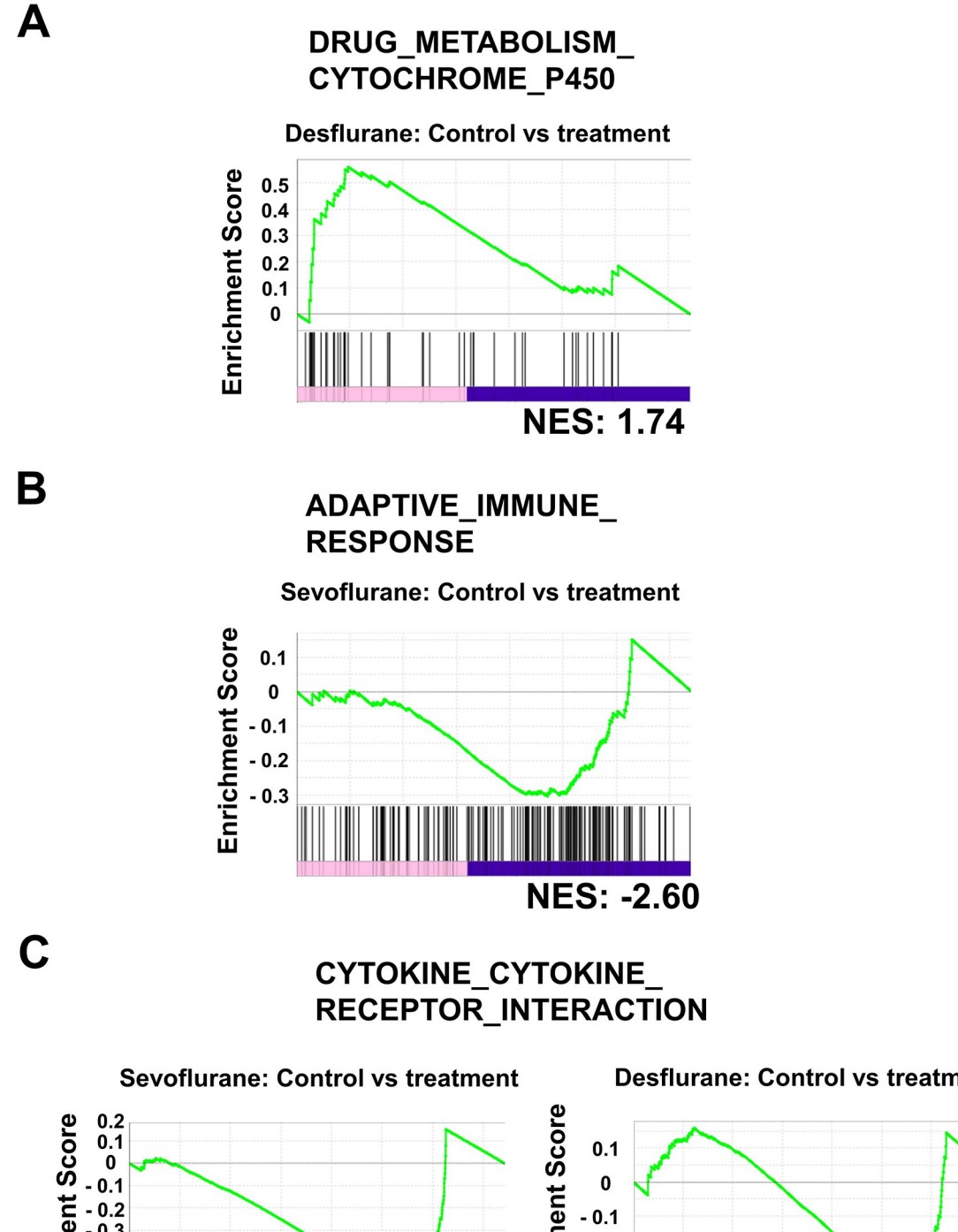

**Fig 3. GSEA.** (A) Snapshot showing a tendency for the positive regulation of genes constituting the term "DRUG_METABOLISM_CYTOCHROME_P450" after treatment with desflurane. Among 47 members of the term included in the list of our RNA-sequence data, 29 and 18 genes were up- and downregulated by desflurane treatment, respectively. A list of genes denoted as leading-edge genes by the analysis is provided in S2A Table. (B) Snapshot showing a tendency for the negative regulation

of genes constituting the term "adaptive immune response" after treatment with sevoflurane. Among 353 members of the term included in the list of our RNA-sequence data, 90 and 263 genes were up- and downregulated by sevoflurane treatment, respectively. The regulatory mode of 21 genes could not be determined because there was no expression in certain samples. A list of leading-edge genes is provided in S2B Table. (C) Snapshots showing a tendency for the negative regulation of genes constituting the term "CYTOKINE_CYTOKINE_RECEPTOR_INTERACTION" after treatment with sevoflurane (left panel) and desflurane (right panel). Among 217 members of the term included in the list of our RNA-sequence data, 132 and 106 genes were downregulated and 50 and 80 genes were upregulated by sevoflurane and desflurane, respectively. The regulatory modes of 35 (for sevoflurane) and 31 (for desflurane) genes could not be determined because there was no expression in certain samples. The lists of genes denoted as leading-edge genes for the treatments of sevoflurane and desflurane are provided in S2C and S2D Table, respectively.

"CYTOKINE_CYTOKINE_RECEPTOR_INTERACTION", which was a commonly identified term in the sevoflurane (left panel) and desflurane (right panel) groups. These two snapshots demonstrated that genes constituting this term were subjected to greater negative regulation by sevoflurane than by desflurane.

## Validation of global gene expression analysis data by the qPCR of representative genes

Next, we conducted qPCR analyses of genes whose expression levels were significantly up- or downregulated by sevoflurane and/or desflurane by means of global gene expression analyses. Specifically, we selected three genes (*Cyp2b1*, *Por, and Alas1*) (Fig 4A) and four genes (*Irf1*, *Mx2*, *Ccl6*, and *Il33*) (Fig 4B) as representative of xenobiotic metabolism and immune response, respectively. First, we confirmed that xenobiotic metabolism-related genes were significantly activated by both inhalation anesthetic agents. Our qPCR data of immune response-related genes also recapitulated the data from the RNA-sequencing analyses. Indeed, our qPCR data confirmed the significant downregulation of the expression of *Irf1* and *Mx2* by sevoflurane and desflurane, which were suggested to be downregulated significantly by both anesthetic agents in the RNA-sequencing analyses. Likewise, our qPCR analyses confirmed the sevoflurane treatment-specific downregulation of the expression of *Ccl6* and *IL33*, which were specifically repressed by sevoflurane, but not desflurane, in the RNA-sequence analyses.

## Discussion

Sevoflurane and desflurane are commonly used inhalational anesthetic agents in modern anesthesia practice [32,33]. Inhalation anesthesia in general is known to induce the activation and repression of genes related to xenobiotic metabolism and immune response, respectively [25–29]. However, because these two halogenated anesthetics have never been compared comprehensively with respect to alterations in the expression levels of genes related to xenobiotic metabolism and immune response, we conducted next-generation sequence analyses using mRNAs from the livers of rats subjected to inhalational anesthesia using sevoflurane or desflurane. Our GO analyses of RNA-sequence data revealed that both anesthetic agents significantly activated numerous genes related to xenobiotic metabolism. These analyses also indicated that desflurane activated these genes to a greater extent than sevoflurane, which was confirmed by GSEA. Given that the magnitude of the activation of xenobiotic metabolism genes parallels the level of protection of the host against xenobiotic-mediated toxicity, these data suggest that a higher level of protection via the xenobiotic metabolizing system might be required in the host when administering desflurane compared with sevoflurane.

Unlike the activated gene sets, no common GO terms were obtained in the analyses of genes downregulated by sevoflurane and desflurane. Although no obvious biological relatedness was apparent among five GO terms obtained in the analyses of downregulated genes by desflurane, we found that most GO terms obtained with sevoflurane were related to immune

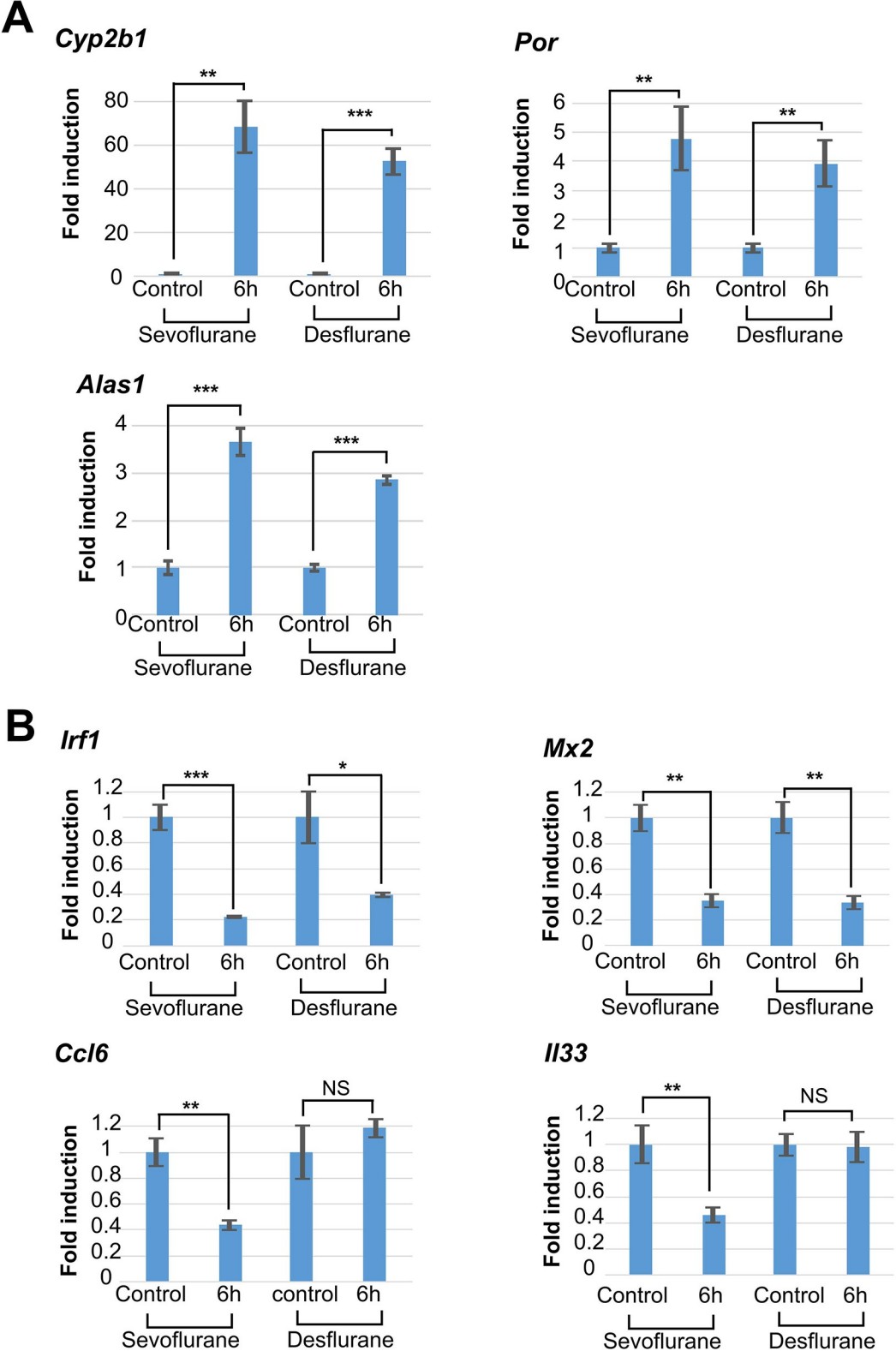

**Fig 4. qPCR analyses of representative genes whose expression was significantly altered by inhalational anesthesia.**
qPCR analyses of the expression of three (A) and four (B) genes as representative genes for xenobiotic metabolism and immune response, respectively. Data from control rats in which livers were recovered immediately after the loss of

consciousness by sevoflurane or desflurane treatment were arbitrarily set to one. Data represent the mean ± SD of three independent experiments. The Student's $t$-test (two-tailed) was conducted to examine statistical significance. ***, $P<0.001$; **, $P<0.01$; *, $P<0.05$; NS, $P>0.05$.

response, indicating that sevoflurane treatment may be strongly linked to immunosuppression. However, a heatmap visualization revealed that desflurane also reduced the expression of immune response genes, albeit less intensively compared with sevoflurane. Our data from GSEA were consistent with the heatmap visualization data, further corroborating the notion that both anesthetic agents repressed immune response-related genes, although sevoflurane exerted a more pronounced repressing effect compared with desflurane.

Multiple studies have demonstrated that volatile anesthetic agents exhibit immunosuppressive effects [27–29]. Therefore, surgeons and anesthesiologists prefer to avoid surgery coupled with general anesthesia for patients vaccinated within the last 2 or 3 weeks because of concerns regarding the insufficient acquisition of immunity by the vaccine. However, because our data suggest that desflurane exerts a less intensive immunosuppressive effect than sevoflurane, our future studies will investigate whether there is a significant difference in the specific immuno-protective ability of rats subjected to inhalation anesthesia with sevoflurane or desflurane after immunization with a vaccine such as that for COVID-19 or influenza virus.

## Supporting information

**S1 Fig. Scatter plot of RNA sequence data.** Upper and lower panels show data obtained from experiments of inhalational anesthesia using sevoflurane and desflurane, respectively. These scatter plots were generated after removing genes whose lengths are equal or shorter than 200 base pairs from the gene list of RNA sequence data. Numerical values shown on the X- and Y-axes are TPM values from RNA sequence data. Genes whose TPM values were increased or decreased more than 2-fold by inhalational anesthesia using sevoflurane or desflurane are indicated as red and blue dots, respectively. The numbers of genes upregulated by 6 hours treatment with sevoflurane and desflurane were 210 and 282, respectively, of which 59 genes overlapped, and 329 and 141 genes were downregulated by sevoflurane and desflurane treatments, respectively, with 31 overlapping genes. S1 Table shows a list of these genes with their official gene symbols.
(PDF)

**S2 Fig. Comparison of the repressing effect on specific gene sets between sevoflurane and desflurane by regression analysis.** (A, B) Coefficient of determination was calculated using the genes in Fig 1D that were upregulated by sevoflurane and/or desflurane more than 2-fold among the members of the GO term "response to xenobiotic stimulus (0009410)" (A, upper panel) and genes shown in Fig 2D that were downregulated by sevoflurane more than 2-fold among the members of the GO term "defense response to virus (0051607)" (B). Lower panel in A shows the result after the removal of *Cyp2b1* gene data as an outlier in the gene set.
(PDF)

**S3 Fig. Comparisons of expression of immune response-related genes between sevoflurane and desflurane treatments.** (A, B) Effect of desflurane treatment on the expressions of genes that contributed to the identification of immune response-related terms as sevoflurane treatment-specific GO terms. Genes downregulated more than 2-fold by sevoflurane treatment were selected among genes constituting the GO terms "immune response (0006952)" (A) and "response to bacterium (0009617)" (B). Relative expression levels in the livers of rats treated with desflurane for 6 hours compared to the control were demonstrated by a heatmap (right

column) along with data obtained by the analyses of livers of rats treated with sevoflurane (left column). Panels shown under each heatmap represent regression analyses for the calculation of the coefficient of determination.
(PDF)

**S4 Fig. GSEA of RNA sequence data from livers of rats subjected to inhalational anesthesia.** (A) A list of gene sets identified by GSEA as significantly activated gene sets by inhalational anesthesia using sevoflurane or desflurane. Three distinct publicly available databases, "biological process of Gene Ontology", "Kyoto Encyclopedia of Genes and Genome", and "Reactome Pathway Database" were used for the analyses, in which the top twenty terms according to their NES values were selected from positively-regulated gene sets with the analyses using each platform, but terms whose *p*-values were greater than 0.05 were eliminated from the list. Terms related to xenobiotic metabolism are shown in light blue. (B) A list of gene sets identified by GSEA as significantly repressed gene sets by inhalational anesthesia using sevoflurane or desflurane. The same criteria used in (A) were used. Commonly identified gene sets by treatment with sevoflurane or desflurane are shown in green. Terms related to immune response are indicated by red font.
(PDF)

**S1 Table. Gene lists that were up- or down-regulated by sevoflurane and/or desflurane.**
(PDF)

**S2 Table. Gene lists denoted as leading-edge genes by GSEA.**
(PDF)

## Acknowledgments

The authors are indebted to Nahomi Nakahara for her technical assistance. The authors also thank Macrogen, Inc. for the RNA-sequencing analyses. We thank J. Ludovic Croxford, PhD, from Edanz (https://jp.edanz.com/ac) for editing a draft of this manuscript.

## Author Contributions

**Conceptualization:** Takehiro Nogi, Tomiei Kazama, Hiroshi Nagasaka, Akihiko Okuda, Tsutomu Mieda.

**Funding acquisition:** Akihiko Okuda.

**Investigation:** Takehiro Nogi, Kousuke Uranishi, Ayumu Suzuki, Masataka Hirasaki, Tina Nakamura, Akihiko Okuda.

**Methodology:** Takehiro Nogi, Kousuke Uranishi, Ayumu Suzuki, Masataka Hirasaki, Tina Nakamura, Tomiei Kazama, Akihiko Okuda, Tsutomu Mieda.

**Project administration:** Akihiko Okuda, Tsutomu Mieda.

**Supervision:** Hiroshi Nagasaka.

**Writing – original draft:** Takehiro Nogi.

**Writing – review & editing:** Tomiei Kazama, Akihiko Okuda, Tsutomu Mieda.

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
