## [Decision Letter · Decision Letter 0]

28 Jun 2023

PONE-D-23-16299Similarity and dissimilarity in alteration of gene expression profile associated with inhalational anesthesia between sevoflurane and desfluranePLOS ONE

Dear Dr. Okuda,

Thank you for submitting your manuscript to PLOS ONE. After careful consideration, we feel that it has merit but does not fully meet PLOS ONE’s publication criteria as it currently stands. Therefore, we invite you to submit a revised version of the manuscript that addresses the points raised during the review process.

The manuscript was reviewed by three reviewers who have specialized research fields in anesthesia, GO analysis, and circadian rhythm. All reviewers expressed the same concern about the time of tissue sampling and a potential difference in tissue sampling time between treated and control rats. Each reviewer also expressed other concerns. I only invite you to submit a revised manuscript if you can address all of the reviewers’ concerns. Redoing the experiments and analysis may be necessary to fully address those concerns (in case you have different sampling times between treated and control rats).

We look forward to receiving your revised manuscript.

Kind regards,

Shin Yamazaki, Ph.D.

Section Editor

PLOS ONE

Journal Requirements:

"The authors are indebted to Nahomi Nakahara for her technical assistance. The authors also thank Macrogen, Inc. for RNA-sequencing analyses and J. Ludovic Croxford, PhD, from Edanz (https://jp.edanz.com/ac) for editing a draft of this manuscript.

This work was supported in part by in-house grants from Saitama Medical University (Internal Grant 21-B-1-07) and Saitama Medical University Hospital (02-E-1-08) to TNo. This work was also supported in part by a grant from the Japan Society for the Promotion of Science KAKENHI (grant number 23H02678) to AO."

"Akihiko Okuda is the recipient of a grant from the Japan Society for the Promotion of Science (KAKENHI: grant number 23H02678). However, the funder had no role in study design, data collection and analysis, decision to publish, or preparation of the manuscript."

Reviewers' comments:

Reviewer's Responses to Questions

**Comments to the Author**

1. Is the manuscript technically sound, and do the data support the conclusions?

Reviewer #1: Partly

Reviewer #2: Yes

Reviewer #3: Partly

2. Has the statistical analysis been performed appropriately and rigorously? 

Reviewer #1: I Don't Know

Reviewer #2: Yes

Reviewer #3: I Don't Know

3. Have the authors made all data underlying the findings in their manuscript fully available?

Reviewer #1: Yes

Reviewer #2: Yes

Reviewer #3: Yes

4. Is the manuscript presented in an intelligible fashion and written in standard English?

Reviewer #1: Yes

Reviewer #2: Yes

Reviewer #3: Yes

5. Review Comments to the Author

Reviewer #1: The authors state that different anesthetic agents (desflurane compared to sevoflurane) have different transcriptomic signatures. They show this by anesthetizing rats for 6 hours with the aforementioned agents and then sacrificing the animals and performing bulk RNA sequencing and qPCR on the tissue of interest, which in this case is the liver. They use briefly induced rats as controls for this study.

The authors further claim that the transcriptomic signature of desflurane and sevoflurane are comparable in their effect on circadian rhythm and drug metabolizing genes and that the only difference was the expression of genes related to cholesterol biosynthesis, which were repressed by sevoflurane, but not by desflurane.

In order to be able to make the statement that circadian rhythm genes are differentially expressed after anesthesia with either sevoflurane or desflurane the authors need to control for circadian timing of the tissue harvesting. The authors do not share any information on harvesting time of the control group samples or the anesthetized samples.

From the methods the impression is made that rats were anesthetized at ZTx for 6hr and sacrificed at ZTx+6, whereas the control group was briefly induced at ZTx and immediately sacrificed. If this is the case, the harvested tissue of the anesthetized rats would be at a different circadian phase, and therefore would have different circadian rhythm gene expression than the controls. However this would be the effect of harvesting time rather than that of the anesthetic used.

In addition this raises the question whether any of the expression changes that the authors see is caused by other oscillating factors, like corticosterone levels for instance. We know that drug metabolism changes throughout the 24-h period, so the difference in gene expression for drug metabolism might also be attributed to the harvesting time. It doesn't seem like the authors have controlled for, or addressed this possible confounder.

Furthermore, the authors do not state the number of animals used for any of their experiments, and this makes it impossible to asses whether the study is well powered or not.

As a minor issue with the availability of sequencing data, the reviewers were not provided a token to access the data, and GEO would not provide one but they said that the publisher would provide one. I didn't follow that up with the journal however.

Reviewer #2: The manuscript described the transcriptome comparison between two inhalational anesthesia agents, sevoflurane and desflurane. From GO analysis and gene set enrichment analysis (GSEA), Nogi et al. identified that they shared similar transcriptome changes. Congruent with previous sevoflurane studies (Gene 2004 Sakamoto et al., Biomed Res. 2009 Nakazato el al.), their RNA-seq study in sevoflurane and desflurane also identified differentially expressed genes (DEGs) in circadian rhythm and drug metabolism pathways. Furthermore, in their downregulated gene dataset, they found that the cholesterol biosynthetic pathway is only responsive to sevoflurane treatment. Since the two anesthesia agents are commonly used and sevoflurane does not tend to induce hepatotoxicity, these transcriptome studies are valuable for understanding the pattern of DEG which causes hepatotoxicity. However, there are some concerns to improve the manuscript.

Major issues:

1. In general, the result section needs to provide more details. Specially, there is no information about the computational and bioinformatic pipelines for RNA-seq data process and DEG identification. Please, provide them in the method section.

2. The RNA sequencing data should not contain small RNA signals because these are depleted during size selection step to remove adaptors. However, there are small RNA signals at Table S1 (e.g. miRNA genes) and Figure 3S (e.g. MIRNA_MEDIATED_GENE_SILENCING_BY_INHIBITION_OF_TRANSLATION). Please, count aligned reads without small RNA genes (<= 200bp) and recalculate TPM values for GO and GSEA analyses. For the analyses, use liver expressed background gene sets that do not include the small RNA genes.

3. In the method section, zeitgeber times are not very clear to collect 0-hour and 6-hour treatment samples. If there is 6-hour gap between the samples, there is additional factor of time that alters the expression of gene sets related to daily circadian rhythms including sevoflurane and desflurane treatments. The phase of the circadian rhythm during treatment with either anesthetic could very likely be the driver of the differential gene expression if there is 6-hour gap between the samples. To reach the same conclusion, the author needs to subtract the rhythmic gene expression changes.

Minor issues

1. Main figures need higher resolution since it is very hard to read the figures.

2. For qPCR experiments, provide if t-tests were one- or two-tailed. Also, note how many samples are used.

3. For Fig. 3, provide more explanation in the figure legend to understand the figure.

4. For Fig. S2B and Fig. 2D, heatmaps showed similar reduced expression at the GO terms. It is better to include plots that show expression correlations with R2 and p-values.

5. For the Venn diagrams at fig 1C and 2C, provide hypergeometric p-values to show how significantly the gene sets are overlapped.

Reviewer #3: This paper concerns differences in gene expression in response to two different inhalation anaesthetic agents (sevoflurane and desflurane). I must admit to being a little confused about the overall point of the study and the results presented. While differences in a number of different gene expression profiles are presented (particularly for genes involved in the molecular mechanics of the circadian clock and in drug metabolism) there were no differences in the expression changes between the two agents. The authors do report a downregulation in the expression of cholersterol biosynthesis genes in response to sevoflurane but I am unsure of the potential clinical significance of this finding.

One significant omission from the manuscript was any description of the timing of the administration of the two agents. Particularly with respect to the analysis of circadian gene expression profiles this is very important. Time of day will have a profound influence on the level of expression of clock genes (and also potentially on the influence of the anaesthetic agents on the expression of clock genes).

A detailed description of the timing and standardization of the administration of GA needs to be including, as does a description of what the implications of different timings might mean in the discussion.

I did not get a clear idea of the point of the manuscript. I think the manuscript would benefit from a substantial re-work in order to make it clear why the authors would expect changes in the gene expression profiles they have seen and what the scientific and clinical relevance of this changes might be.

Other more minor comments include

Introduction

1. GA is not sedation.

2. Isoflurane is the most commonly used inhalational agent in the world not sevoflurane

6. PLOS authors have the option to publish the peer review history of their article (what does this mean?). If published, this will include your full peer review and any attached files.

Reviewer #1: No

Reviewer #2: **Yes: **Chang Hoon Lee

Reviewer #3: No

---

## [Author Response · Author response to Decision Letter 0]

6 Nov 2023

Response to Reviewers

Reviewer #1: 

First comment from reviewer #1

In order to be able to make the statement that circadian rhythm genes are differentially expressed after anesthesia with either sevoflurane or desflurane the authors need to control for circadian timing of the tissue harvesting. The authors do not share any information on harvesting time of the control group samples or the anesthetized samples.

From the methods the impression is made that rats were anesthetized at ZTx for 6hr and sacrificed at ZTx+6, whereas the control group was briefly induced at ZTx and immediately sacrificed. If this is the case, the harvested tissue of the anesthetized rats would be at a different circadian phase, and therefore would have different circadian rhythm gene expression than the controls. However this would be the effect of harvesting time rather than that of the anesthetic used.

As this reviewer pointed out, we had harvested livers from control and experimental rats at 9 am and 3 pm, respectively. Therefore, we conducted the same experiments in which all livers, including those in the control group, were harvested at 3 pm. Then, we compared the expression levels of circadian genes by quantitative PCR using RNAs prepared from the newly harvested liver samples. We found minimal differences in the expression levels between liver RNAs from control and anesthetized rats, indicating that the marked activation of circadian rhythm-related genes we observed previously were not due to inhalation anesthesia but rather the difference in their circadian rhythm phase. Because of this finding, we repeated the analyses using data from the newly conducted RNA-sequencing to revise our manuscript.

Second comment from reviewer #1

In addition, this raises the question whether any of the expression changes that the authors see is caused by other oscillating factors, like corticosterone levels for instance. We know that drug metabolism changes throughout the 24-h period, so the difference in gene expression for drug metabolism might also be attributed to the harvesting time. It doesn't seem like the authors have controlled for, or addressed this possible confounder.

As this reviewer stated, we found that not only canonical circadian genes but also a substantial number of genes including those for xenobiotic metabolism such as Alas1 and Por were substantially affected by the timing of tissue harvesting. However, unlike circadian genes, the anesthetic procedure-dependent activation of these genes was still evident in the newly conducted analyses, albeit to a lower level than before.

Third comment from reviewer #1

Furthermore, the authors do not state the number of animals used for any of their experiments, and this makes it impossible to assess whether the study is well powered or not.

Although N=1 for the RNA-seq data, quantitative PCR used to validate the RNA-seq data were conducted with N=3. We described this explicitly in Methods section in the revised text (page 6 lines 16 to 18 for quantitative PCR and page 7 lines 3 to 5 for RNA-seq data).

Fourth comment from reviewer #1

As a minor issue with the availability of sequencing data, the reviewers were not provided a token to access the data, and GEO would not provide one but they said that the publisher would provide one. I didn't follow that up with the journal however.

We must apologize for this issue. We did not know that we had to create and provide a token so that any reviewers can access the private data. We have rectified this and provide a token (snqnkssarfwjjip) for our RNA sequence data (GSE244436). Once again, we would like to apologize any inconvenience that this reviewer encountered because of this issue.

 

Reviewer #2: 

First comment from reviewer #2

In general, the result section needs to provide more details. Specially, there is no information about the computational and bioinformatic pipelines for RNA-seq data process and DEG identification. Please, provide them in the method section.

According to this reviewer’s comment, we provided the details of pipelines for the RNA-seq data process and DEG identification in the Methods section (page 7 line 1 to page 8 line 4). We also added information about the timing of tissue harvesting in the Methods section (page 5 line 18 to page 6 line 2)

Second comment from reviewer #2

The RNA sequencing data should not contain small RNA signals because these are depleted during size selection step to remove adaptors. However, there are small RNA signals at Table S1 (e.g. miRNA genes) and Figure 3S (e.g. MIRNA_MEDIATED_GENE_SILENCING_BY_INHIBITION_OF_TRANSLATION). Please, count aligned reads without small RNA genes (<= 200bp) and recalculate TPM values for GO and GSEA analyses. For the analyses, use liver expressed background gene sets that do not include the small RNA genes.

Because we did not know how to delete the data of the small RNA genes (≤200 bp), we removed all non-coding RNAs from the list including longer RNAs (>200 bp) from the list using BioMart web tool (https://asia.ensembl.org/info/data/biomart/index.html) as described on page 7 lines 14 to 15 and conducted GO and GSEA analyses in response to this reviewer’s comment. We hope that these procedures are satisfactory for this reviewer.

Third comment from reviewer #2

In the method section, zeitgeber times are not very clear to collect 0-hour and 6-hour treatment samples. If there is 6-hour gap between the samples, there is additional factor of time that alters the expression of gene sets related to daily circadian rhythms including sevoflurane and desflurane treatments. The phase of the circadian rhythm during treatment with either anesthetic could very likely be the driver of the differential gene expression if there is 6-hour gap between the samples. To reach the same conclusion, the author needs to subtract the rhythmic gene expression changes.

As this reviewer and other reviewers pointed out, there was a 6-hour gap in liver harvesting time between the control and experimental rats. Therefore, we conducted the same experiments in which all livers, including those of the control group, were harvested at the same time, at 3 pm. Then, we compared the expression levels of circadian genes by quantitative PCR using RNAs prepared from newly harvested liver samples and found only minimal differences in their expression levels between liver RNAs from anesthetized rats and control rats. This indicated that the marked activation of circadian rhythm-related genes we observed previously was not due to inhalation anesthesia but rather to the difference in their circadian rhythm phase. Because of this finding, we repeated the analyses using data from newly conducted RNA-sequencing to revise our manuscript.

Fourth comment from reviewer #2

Main figures need higher resolution since it is very hard to read the figures.

In the revised manuscript, we provided figures with a high resolution (600 dot per inch). We hope that the resolution of the newly provided files is satisfactory.

Fifth comment from reviewer #2

For qPCR experiments, provide if t-tests were one- or two-tailed. Also, note how many samples are used.

A two-tailed t-test (N=3) was conducted for all experiments. We described this in Methods section (page 8 lines 15 to 17) as well as in legends of Figure 4 (page 34 lines 4 to 6) in the revised text.

Sixth comment from reviewer #2

For Fig. 3, provide more explanation in the figure legend to understand the figure.

We have provided information related to the number of genes included in each gene set and how many genes were activated or repressed. We hope that our response is satisfactory.

Seventh comment from reviewer #2

For Fig. S2B and Fig. 2D, heatmaps showed similar reduced expression at the GO terms. It is better to include plots that show expression correlations with R2 and p-values.

We provided these data in Fig. S2 and S3 in the revised manuscript. These regression analyses revealed that neither xenobiotic metabolism-related genes nor immune response-related genes had a noticeable correlation related to variations in their expression levels between the sevoflurane and desflurane groups. We described about these results in the text (page 11 lines 1 to 7, pages 13 lines 5 to 11 and page 18 lines 10 to 17).

Eighth comment from reviewer #2

For the Venn diagrams at fig 1C and 2C, provide hypergeometric p-values to show how significantly the gene sets are overlapped.

As suggested, we provided these data in Fig. 1A and Fig. 2A in the revised manuscript.

 

Reviewer #3: 

First comment from reviewer #3

This paper concerns differences in gene expression in response to two different inhalation anaesthetic agents (sevoflurane and desflurane). I must admit to being a little confused about the overall point of the study and the results presented. While differences in a number of different gene expression profiles are presented (particularly for genes involved in the molecular mechanics of the circadian clock and in drug metabolism) there were no differences in the expression changes between the two agents. The authors do report a downregulation in the expression of cholersterol biosynthesis genes in response to sevoflurane but I am unsure of the potential clinical significance of this finding. One significant omission from the manuscript was any description of the timing of the administration of the two agents. Particularly with respect to the analysis of circadian gene expression profiles this is very important. Time of day will have a profound influence on the level of expression of clock genes (and also potentially on the influence of the anaesthetic agents on the expression of clock genes).

As this reviewer and other reviewers pointed out, there was a 6-hour gap in liver harvesting time between the control and experimental rats. Therefore, we conducted the same experiments in which all livers, including those of the control group, were harvested at the same time, at 3 pm. Then, we compared the expression levels of circadian genes by quantitative PCR using RNAs prepared from newly harvested liver samples and found only minimal differences in their expression levels between liver RNAs from anesthetized rats and control rats. This indicated that the marked activation of circadian rhythm-related genes we observed previously was not due to inhalation anesthesia but rather to the difference in their circadian rhythm phase. Because of this finding, we repeated the analyses using data from newly conducted RNA-sequencing. 

GO analyses using data from the newly conducted RNA-sequencing revealed that terms related to circadian rhythm were no longer significant terms with sevoflurane-treated sample, although the term “circadian regulation of gene expression (GO:0032922)” was identified with desflurane-treated sample in the 12th position from the top. Unexpectedly, the GO term “cholesterol biosynthetic process (0006695)” was not identified with samples from rats treated with sevoflurane, indicating that a 6-hour gap in tissue harvesting not only affected canonical circadian genes in their expression levels but also affected many other genes whose expressions oscillate between day and night. These findings indicate that adjusting the timing of tissue harvesting within the day are critical when comparing control and experimental animals.

We would like to thank this reviewer and other reviewers because we would not have conducted the RNA-sequencing analyses again without these invaluable comments.

Second comment from reviewer #3

I did not get a clear idea of the point of the manuscript. I think the manuscript would benefit from a substantial re-work in order to make it clear why the authors would expect changes in the gene expression profiles they have seen and what the scientific and clinical relevance of this changes might be.

We have intensively edited the text, especially in the Abstract (page 2 lines 2 to 9) and Introduction (page 4 lines 3 to 9) in response to this reviewer’s comment. We edited the text with a focus on the comparison of the effects of sevoflurane and desflurane on the expressions of genes related to xenobiotic metabolism and immune response, because no comprehensive comparison has been reported to date. We hope the newly edited text is satisfactory.

Third comment from reviewer #3

GA is not sedation.

To avoid confusion, we eliminated this expression from the text.

Fourth comment from reviewer #3

Isoflurane is the most commonly used inhalational agent in the world not sevoflurane

As suggested, we changed the expression from “the most---” to “one of the most---” (first line in the abstract and second sentence in the Introduction section).

---

## [Decision Letter · Decision Letter 1]

18 Dec 2023

PONE-D-23-16299R1Similarity and dissimilarity in alterations of the gene expression profile associated with inhalational anesthesia between sevoflurane and desfluranePLOS ONE

Dear Dr. Okuda,

Thank you for submitting your manuscript to PLOS ONE. After careful consideration, we feel that it has merit but does not fully meet PLOS ONE’s publication criteria as it currently stands. Therefore, we invite you to submit a revised version of the manuscript that addresses the points raised during the review process.

We look forward to receiving your revised manuscript.

Kind regards,

Shin Yamazaki, Ph.D.

Section Editor

PLOS ONE

**Additional Editor Comments:**

First, I apologize for taking a significant amount of time to evaluate your amended manuscript. Your amended manuscript was reviewed by two reviewers who also reviewed your original manuscript. Both reviewers have several important suggestions. Please revise the manuscript accordingly. To address the comments by reviewer #1, extensive manuscript revision and language editing are necessary. I believe this will make your manuscript legible, therefore. I strongly encourage you to do so. PLoS ONE doesn’t perform language editing, so the accepted manuscript will be published as is. After you amend the manuscript, please use a scientific language editing service. When you submit, please provide the service you have used.

Reviewers' comments:

Reviewer's Responses to Questions

**Comments to the Author**

1. If the authors have adequately addressed your comments raised in a previous round of review and you feel that this manuscript is now acceptable for publication, you may indicate that here to bypass the “Comments to the Author” section, enter your conflict of interest statement in the “Confidential to Editor” section, and submit your "Accept" recommendation.

Reviewer #1: All comments have been addressed

Reviewer #2: (No Response)

2. Is the manuscript technically sound, and do the data support the conclusions?

Reviewer #1: Yes

Reviewer #2: Partly

3. Has the statistical analysis been performed appropriately and rigorously? 

Reviewer #1: Yes

Reviewer #2: Yes

4. Have the authors made all data underlying the findings in their manuscript fully available?

Reviewer #1: Yes

Reviewer #2: Yes

5. Is the manuscript presented in an intelligible fashion and written in standard English?

Reviewer #1: No

Reviewer #2: Yes

6. Review Comments to the Author

Reviewer #1: The authors compared two different volatile anesthetics (sevoflurane and desflurane) and compared the transcriptomic response to these anesthetic by next-gen sequencing of the rat liver. Their findings suggest that the differentially expressed genes of these anesthetics have different gene ontology profiles, and that the profile belonging to sevoflurane seems to be suggestive of a decreased immune response profile. Compared to the last submission of this paper, the findings have really changed, and I think this paper is a better representation of what is happening in the liver after anesthesia with sevoflurane or desflurane. Even though I believe the science is sound and the findings and conclusions are now in line with the experimental setup, the way it is presented is very difficult to follow. Below I have highlighted some segments that are examples, but the manuscript is overwhelming with these kind of sentencing constructs. Based on the difficulty of following the manuscript I fear that the interesting findings will be lost to confusion.

example 1: this following segment is one sentence.

"In accordance with these data, regression analysis demonstrated no apparent correlation in alterations of the expressions of these genes between sevoflurane and desflurane treatments in case that data of Cyp2b1 expression were removed as an outlier in the gene set (S2A Fig, lower panel), although an analysis of all genes shown in Figure 1D suggests a high correlation (R2=0.9371) because of the marked activation of Cyp2b1 gene expression by both anesthetic agents (S2A Fig, upper panel)."

They are saying that the expression patterns between the two modalities (sevoflurane vs desflurane) show now apparent correlation when the expression of Cyp2b1 is excluded from the analysis. Apparently the expression of Cyp2b1 is an outlier that skews the correlation analysis significantly.

example 2: this following segment is one sentence.

"Although no GO term related to immunological response was obtained in the analyses of genes downregulated by desflurane, we inspected the gene expression levels in the livers of rats subjected to inhalational anesthesia with desflurane as to genes that were contributed to the identification of the GO term, “defense response to virus” (GO:0051607) as sevoflurane- specific terms because of their downregulation more than 2-fold by sevoflurane treatment."

I think it's clear that they wanted to compare the expression levels of a gene set from a sevoflurane specific GO term between the two different treatments, but the wording does not make that easy to understand.

Example 3: this following segment is one sentence.

Regression analyses revealed that, although there was a weak correlation between genes that contributed to the identification of the GO term, “response to bacterium” (GO:0009617) as a sevoflurane treatment-specific term (R2=0.2605) (S3B Fig), no appreciable statistical significance was apparent with other gene sets (genes downregulated more than 2-fold by sevoflurane treatment among genes comprising the terms of GO:0051607 and GO:0006955) (S2B Fig and S3A Fig).

I had a very difficult time following what the authors were trying to say here.

Reviewer #2: The revised manuscript has appropriately addressed most of my comments. However, there is one aspect where the authors need to make a correction in their approach.

To delete the data of the small RNA genes (≤200 bp), removing all non-coding RNAs is not the optimal approach. While it effectively eliminates most small RNAs, it could also lead to the depletion of all long non-coding RNAs, remaining some small RNAs originate from protein-coding genes.

To identify small RNA genes (≤200bp), you can utilize a GTF file from Ensembl that aligns with your reference genome. After calculating transcript lengths, filter out all small transcripts (≤200bp) from the GTF file. Using the refined GTF file, you can get a list of genes that transcribe RNAs ≥200bp. Consequently, you can accurately count your aligned reads, excluding small RNAs (≤200bp).

7. PLOS authors have the option to publish the peer review history of their article (what does this mean?). If published, this will include your full peer review and any attached files.

Reviewer #1: No

Reviewer #2: No

---

## [Author Response · Author response to Decision Letter 1]

9 Jan 2024

Response to Reviewers

Reviewer #1: 

First comment from reviewer #1

The way it is presented is very difficult to follow. Below I have highlighted some segments that are examples. 

example 1: this following segment is one sentence.

"In accordance with these data, regression analysis demonstrated no apparent correlation in alterations of the expressions of these genes between sevoflurane and desflurane treatments in case that data of Cyp2b1 expression were removed as an outlier in the gene set (S2A Fig, lower panel), although an analysis of all genes shown in Figure 1D suggests a high correlation (R2=0.9371) because of the marked activation of Cyp2b1 gene expression by both anesthetic agents (S2A Fig, upper panel).

We agree the English in this section was extremely difficult to follow as it was difficult to explain the data explicitly, because the regression analyses conducted according to a comment from another reviewer with a gene set shown in Figure 1D apparently contradicted the result obtained from the heatmap visualization. However, in the revised manuscript, this section of the text was edited extensively with the aid of professional English editing service so that it can be understood more easily (page 10 lines 1 to 8). We hope that newly edited text is satisfactory.

example 2: this following segment is one sentence.

"Although no GO term related to immunological response was obtained in the analyses of genes downregulated by desflurane, we inspected the gene expression levels in the livers of rats subjected to inhalational anesthesia with desflurane as to genes that were contributed to the identification of the GO term, “defense response to virus” (GO:0051607) as sevoflurane- specific terms because of their downregulation more than 2-fold by sevoflurane treatment."

Again, we agree this section was extremely difficult to follow. We did not use an entire gene set constituting the GO term “defense response to virus”, but rather selected genes with expression levels reduced more than 2-fold by sevoflurane treatment from the gene set. Accordingly, we have rewritten the text in the revised manuscript with the aid of a professional editing service (page 12 lines 10 to 16) and we hope this is now easier to understand.

Example 3: this following segment is one sentence.

Regression analyses revealed that, although there was a weak correlation between genes that contributed to the identification of the GO term, “response to bacterium” (GO:0009617) as a sevoflurane treatment-specific term (R2=0.2605) (S3B Fig), no appreciable statistical significance was apparent with other gene sets (genes downregulated more than 2-fold by sevoflurane treatment among genes comprising the terms of GO:0051607 and GO:0006955) (S2B Fig and S3A Fig).

We also edited this portion extensively so that it is easier to understand (page 13 lines 7 to 15). We hope that newly edited text is satisfactory for this reviewer.

Second comment from reviewer #1

The manuscript is overwhelming with these kind of sentencing constructs. 

In the revised manuscript, we tried to avoid making overstatements. In particular, we removed the following sentences from the Discussion section, because we felt that they were too exaggerated and could be removed without affecting the scientific content.

“This finding might alter the current concept that one of major side-effects of general anesthesia with a volatile anesthetic agent is immunosuppression.”

Reviewer #2: 

Comment from reviewer #2

The revised manuscript has appropriately addressed most of my comments. However, there is one aspect where the authors need to make a correction in their approach.

To delete the data of the small RNA genes (≤200 bp), removing all non-coding RNAs is not the optimal approach. While it effectively eliminates most small RNAs, it could also lead to the depletion of all long non-coding RNAs, remaining some small RNAs originate from protein-coding genes.

According to this reviewer’s comment, we removed the small RNA genes (≤200 bp) from the gene list instead of removing all the non-coding genes, and re-conducted all the analyses shown in Figures 1-3 and all the supplementary figures and tables.

---

## [Decision Letter · Decision Letter 2]

23 Jan 2024

Similarity and dissimilarity in alterations of the gene expression profile associated with inhalational anesthesia between sevoflurane and desflurane

PONE-D-23-16299R2

Dear Dr. Okuda,

We’re pleased to inform you that your manuscript has been judged scientifically suitable for publication and will be formally accepted for publication once it meets all outstanding technical requirements.

Kind regards,

Shin Yamazaki, Ph.D.

Section Editor

PLOS ONE

Reviewers' comments:

Reviewer's Responses to Questions

**Comments to the Author**

1. If the authors have adequately addressed your comments raised in a previous round of review and you feel that this manuscript is now acceptable for publication, you may indicate that here to bypass the “Comments to the Author” section, enter your conflict of interest statement in the “Confidential to Editor” section, and submit your "Accept" recommendation.

Reviewer #1: All comments have been addressed

Reviewer #2: All comments have been addressed

2. Is the manuscript technically sound, and do the data support the conclusions?

Reviewer #1: Yes

Reviewer #2: Yes

3. Has the statistical analysis been performed appropriately and rigorously? 

Reviewer #1: Yes

Reviewer #2: Yes

4. Have the authors made all data underlying the findings in their manuscript fully available?

Reviewer #1: Yes

Reviewer #2: Yes

5. Is the manuscript presented in an intelligible fashion and written in standard English?

Reviewer #1: Yes

Reviewer #2: Yes

6. Review Comments to the Author

Reviewer #1: Between the first and second resubmission of this paper I feel the authors have addressed all my previous comments in a acceptable fashion. And I will be happy to support the submission of this paper. I think it adds interesting new insight in the effects of inhalation anesthetics in peripheral organs.

Reviewer #2: In the revised manuscript, the authors has appropriately addressed all my comments as reanalyzing data.

7. PLOS authors have the option to publish the peer review history of their article (what does this mean?). If published, this will include your full peer review and any attached files.

Reviewer #1: No

Reviewer #2: No

---

## [Editor Report · Acceptance letter]

19 Mar 2024

PONE-D-23-16299R2 

PLOS ONE

Dear Dr. Okuda, 

I'm pleased to inform you that your manuscript has been deemed suitable for publication in PLOS ONE. Congratulations! Your manuscript is now being handed over to our production team.

Kind regards, 

on behalf of

Dr. Shin Yamazaki 

Section Editor

PLOS ONE